# Towards personalised anti-microbial and immune approaches to infections in acute care. Can real-time genomic-informed diagnosis of pathogens, and immune-focused therapies improve outcomes for patients? An observational, experimental study protocol

Samuel Quarton[1,2]*, Kirsty McGee[1,2], Nicola Cumley[1,3], Mahboobeh Behruznia[1,3], Charlotte Jeff[1], Kylie Belchamber[1,2], Michael Cox[1,3], David Thickett[1,2], Aaron Scott[1,2], Dhruv Parekh[1,2], Alan McNally[1,3ↇ], Elizabeth Sapey[1,4,5ↇ]

1 Birmingham Biomedical Research Centre, University Hospitals Birmingham NHS Foundation Trust, Birmingham, United Kingdom, 2 Institute of Inflammation & Ageing, College of Medical and Dental Sciences, University of Birmingham, Birmingham, United Kingdom, 3 Institute of Microbiology & Infection, College of Medical and Dental Sciences, University of Birmingham, Birmingham, United Kingdom, 4 Director of PIONEER: Health Data Research UK (HDRUK) Health Data Research Hub for Acute Care, Institute of Inflammation and Ageing, University of Birmingham, Birmingham, United Kingdom, 5 Patient Safety Research Collaborative, University Hospitals Birmingham NHS Foundation Trust, Birmingham, United Kingdom

ↇ These authors contributed equally to this work.
* s.quarton@bham.ac.uk

## Abstract

## Introduction

Infection causes a vast burden of disease, with significant mortality, morbidity and costs to health-care systems. However, identifying the pathogen causative infection can be challenging, resulting in high use of broad-spectrum antibiotics, much of which may be inappropriate. Novel metagenomic methods have potential to rapidly identify pathogens, however their clinical utility for many infections is currently unclear. Outcome from infection is also impacted by the effectiveness of immune responses, which can be impaired by age, co-morbidity and the infection itself. The aims of this study are twofold:

1. To compare diversity of organisms identified and time-to-result using metagenomic methods versus traditional culture-based techniques, to explore the potential clinical role of metagenomic approaches to pathogen identification in a range of infections.

2. To characterise the *ex vivo* function of immune cells from patients with acute infection, exploring host and pathogen-specific factors which may affect immune function and overall outcomes.

**Data Availability Statement:** All data are within the paper.

**Funding:** This study is funded by the National Institute for Health and Care Research (NIHR) Birmingham Biomedical Research Centre (BRC). The views expressed are those of the author(s) and not necessarily those of the NIHR or the Department of Health and Social Care. It is also supported by PIONEER, the Health Data Research UK (HDR-UK) Health Data Research Hub in acute care. HDR-UK is an initiative funded by the UK Research and Innovation, Department of Health and Social Care (England) and the devolved administrations, and leading medical research charities. The funders had no role in study design, data collection and analysis, decision to publish, or preparation of the manuscript.

**Competing interests:** Elizabeth Sapey needs to declare funding from HDR-UK, The Wellcome Trust, MRC, NIHR, and Asthma + Lung UK. These fundings are not commercial but charitable and do not relate to the study but to Elizabeth Sapey. Alan McNally reports funding from UK Research and Innovation and Wellcome Trust. No other authors have interests to declare.

## Methods

This is a prospective observational study of patients with acute infection. Patients with symptoms suggestive of an acute infection will be recruited, and blood and bodily fluid relevant to the site of infection collected (for example, sputum and naso-oropharyngeal swabs for respiratory tract infections, or urine for a suspected urinary tract infection). Metagenomic analysis of samples will be compared to traditional microbiology, alongside the antimicrobials received. Blood and respiratory samples such as bronchoalveolar lavage will be used to isolate immune cells and interrogate immune cell function. Where possible, similar samples will be collected from matched participants without a suspected infection to determine the impact of infection on both microbiome and immune cell function.

## Introduction

Acute infection is responsible for a vast and increasing burden on healthcare systems with hospitalisations in England and Wales due to infections increasing more than 412% in the past 20 years [1]. Furthermore, hospitalizations due to infection are predicted to double by 2040 as a result of our ageing population [2].

The World Health Organisation estimate there are more than eleven million deaths from sepsis alone each year [3]. Many patients who survive infection experience significant short and long-term impacts, including increased functional impairment in activities of daily living, mild-moderate cognitive impairment and substantial depressive symptoms [4]. Readmission to hospital after a severe infection is common [5] and these patients experience an increased mortality for at least five years beyond hospital discharge [6]. Despite this, treatments for many infectious diseases in acute care pathways have not changed for decades.

### Infections, their causes and treatments

There is strong evidence that early, appropriate treatment with anti-microbial therapies reduce the mortality, complications and recovery time from infections [7]. However, it is still challenging to identify if a patient has an infection, and what type of organism may be causing it. In over half of the cases of hospitalised infections, the causal bacteria, virus or fungus is not identified, resulting in empirical use of broad-spectrum antibiotics based on the most likely causative pathogens and local patterns of antimicrobial resistance (AMR) [8]. The Centers for Disease Control and Prevention (CDC) in the USA estimate that up to 50% of all antibiotics prescribed for people are either not needed or are inappropriate [9].

AMR is an increasing global concern and inappropriate antibiotic use has been highlighted as a key driver [10]. For individual patients, adverse drug events (ADE) from antibiotics are common, affecting approximately 20% of hospitalised patients who receive antibiotics [11]. However, withholding antibiotics while awaiting definitive test results is also a risk for patients. A recent study of GP records in the UK suggested that practices with the lowest prescribing rates of antibiotics were associated with a higher rate of hospitalised infections [12].

Ideally, microbiology laboratories would quickly and accurately identify whether a bacteria, virus, fungus or protozoa is causing an infection and provide information about the AMR patterns of the pathogen, so the correct antimicrobial (or indeed no antimicrobial) is given. However, current techniques often contribute to significant delay between sample collection and report read out, termed laboratory turnaround time (LTAT), with an average LTAT of three

days being reported for culture-based approaches in an inpatient population [13]. While some of this delay will be due to common factors such as the efficiency of clinical teams and hospital systems, the time taken for traditional microbial culture is also a significant component. Other assays such as antigen-based testing can provide quicker results but require selective testing and are not available for all pathogens.

Modern genomic and molecular-based methods have the potential to rapidly identify not only the exact type of pathogen, but also genetic mutations which might affect virulence [14] and antibiotic resistance [15, 16]. The pathogen-agnostic nature of metagenomic next-generation sequencing (mNGS) can also allow for the detection of rare or emerging pathogens [17, 18]. However, how best to interpret the clinical significance of positive results is uncertain given organisms identified may represent colonisation rather than infection [16]. It is also currently unclear whether these cutting-edge techniques can be applied across all common infections in hospitals or community settings and be embedded into a clinical service.

Even with appropriate antimicrobials, a patient must mount an effective immune response to the invading pathogen. However, factors including age and certain co-morbidities can render this complex system ineffective and even damaging to the host, as demonstrated with neutrophil function in age, frailty and some non-communicable diseases [19–21].

Profound and prolonged neutrophil dysfunction has also been described in response to severe infection. In older adults with infection the accuracy of neutrophil migration and targeting of bacteria is impaired for at least 6 weeks in survivors [22]. The mechanisms by which these altered immune responses occur is yet to be fully characterised. Identifying these may enable the development of treatments that improve immune cell functions, thereby improving patient outcomes. More work is needed to increase our understanding of the potential mechanisms of blunted immune responses, including whether different pathogens impact immune function differently.

## Rationale

Interventions that increase the early identification of infections, enable the right antibiotic to be prescribed where needed, reduce reliance on broad spectrum antibiotics and improve how the immune system fights infections could have significant impact upon survival, patient quality of life, complications from infections, NHS costs (including readmissions) and AMR patterns.

Our overarching aim is to improve outcomes from infections and associated poor immune responses, through state-of-the-art real-time diagnostics of pathogens, a better understanding of infection-pathogen/host interactions which inhibit the immune system and by identifying novel ways to improve the immune response. This exploratory study forms the first steps towards this aim, seeking to assess whether it is possible to improve the identification of likely causative pathogens during infections using metagenomic assessment of potentially infected bodily fluids. Further, this study will assess whether there are changes in immune cell function (with a focus on innate immunity) during infections in general, and whether specific infections or just the severity of any infection are associated with an exaggerated state of immune dysfunction.

## Builds on previous work

- Causative pathogens are currently not identified in over half of patients hospitalised with infection

- Up to 50% of antibiotic prescriptions are estimated to be inappropriate

- Metagenomic next generation sequencing techniques can identify organisms with high sensitivity

- Severe infection is known to cause prolonged immune dysfunction in survivors

### Differs from previous work

- Combination of microbiological and clinical data together with immune cell function in one study, allowing for analysis into the interplay of these factors.

Broad scope investigating all acute infection. This will allow for differences depending on infection source to be explored.

### Study hypotheses and outcomes

1. Real-time, genomics-informed diagnosis of pathogens and antimicrobial sensitivities can improve the rates of pathogen identification, and may lead to reduced reliance on broad-spectrum antibiotics.

2. Molecular and immunological interactions between pathogen and host modulate the immune response. Understanding these pathways may identify novel therapeutic targets.

Specific outcome measures are identified in Table 1, below.

## Methods

### Study design

This is an observational, experimental medicine study using samples taken from human participants recruited from hospital and community settings. Patients will initially be recruited from four hospital sites in Birmingham, UK, with recruitment starting in October 2023 until October 2027.

Age and co-morbidity matched controls without infection will also be recruited. Full inclusion and exclusion criteria are listed in Table 2 below. After discussion with pregnant women and their partners, pregnant women will be included in this study as there is evidence of poorer outcomes from infection in pregnant women [23]. Here, samples will only be collected where they are minimally invasive (for example a blood test) or non-invasive (for example, a skin swab or urine sample).

### Participant recruitment

Potential participants will be screened for by the members of the direct care team following an admission or medical consultation for a presumed infection. Informed consent will be sought from all participants where they have capacity.

**Table 1. Primary and secondary outcomes.**

| Co-primary outcomes: | Proportion of patients in whom a pathogen is identified.<br>Neutrophil chemotaxis in patients with acute infection. |
|---|---|
| Secondary outcomes: | Turn-around times for microbiological results<br>Genotypic resistance profiles of identified organisms, and comparison to antimicrobials given.<br>Other neutrophil functional and signalling parameters including neutrophil phagocytosis, neutrophil extracellular trap formation, neutrophil phenotype and transcriptomics.<br>30 day, 90 day and 1 year mortality |

**Table 2. Inclusion and exclusion criteria.**

| Infection Cohort | |
|---|---|
| **Inclusion Criteria** | **Exclusion criteria** |
| Age ≥18 years<br> • Patient with a confirmed or suspected diagnosis of infection (any organ or cause) which has either<br> ○ Triggered an acute care contact (admission to hospital or community healthcare review).<br> ○ Occurred during a hospital admission<br> Consent is gained within 48 hours of clinical diagnosis of infection | Patient declines consent<br>Personal Consultee, when available, does not provide consent<br>Professional Consultee, if used, does not provide consent<br>Patient treatment to be palliative in nature<br>Patient does not meet the inclusion criteria<br>Patient already enrolled in an interventional research study of a novel / unlicensed drug / therapy (patients enrolled in interventional studies examining the clinical application or therapeutic effects of widely accepted, "standard" treatments are not excluded) |
| **Control Cohort** | |
| **Inclusion Criteria** | **Exclusion criteria** |
| Age and morbidity matched control with either:<br> • No self-reported infective episodes in past 3 months<br>OR<br> • In hospital with no suspicion of or confirmed infection | As above |

In time, we wish to establish if rapid microbial genomics could be used in the community to improve antibiotic stewardship for GP practices. Therefore, although initially we will recruit all patients from hospitals, during this programme of work we will work with long-term care facilities, community health centres and GPs to test how quickly we can process samples from the community. All studies processes would remain the same. The additional participating sites will be included by non-substantial amendment at the appropriate time.

Controls for the study will be recruited from the "Birmingham 1000 Elders" database or from healthy volunteers recruited through adverts. The 1000 Elders cohort is a group of older adults who have voluntarily agreed to be contacted to participate in research studies at the University of Birmingham. At the time of recruitment, they will have no active signs of infection or inflammation, with clinical observations within the normal physiological range. Controls will also include people admitted to hospital where an infection is not suspected. Many acute illnesses are associated with inflammation which could alter immune cell function. This group will help us understand if changes seen to immune cell function are due to the inflammation or the presence of the pathogen.

**Patients lacking capacity.** Some patients who would be suitable for the study may not have capacity to consent at the time of recruitment due to altered level of consciousness or heavy sedation facilitating invasive ventilation. It was felt important to include these patients to ensure innovation benefits this particularly underserved group. In cases where capacity is lacking, the patient's relative, friend or partner will be approached in the first instance as a personal consultee to seek to establish consent. In the event that there is no identifiable relative, friend or partner available, a doctor (Consultant) from the patient's direct care team who is unconnected to the study will be asked to be the patient's Professional Consultee. If and when the patient regains capacity, retrospective consent will be sought during their hospital admission. If the patient does not wish to take part in the study, they will have the choice as to whether collected data and samples are used or destroyed.

The study team would be careful to avoid distress and would not take samples from people unless they were cooperative with sample collection.

## Sample size

Power calculations have been based on co-primary outcomes of the proportion of patients with infection in whom a pathogen is identified, and extent of neutrophil chemotaxis.

Power calculations were based on community acquired pneumonia (CAP) and urinary tract infection, as two common acute presentations. In patients with CAP, a recent systematic review of aetiology suggested an organism is identified by standard methods (sputum and blood culture) in 40% of cases. This improved to 49.8% where viral PCR is also performed [24]. In studies of severe pneumonia, metagenomic next-generation sequencing (mNGS) improved microbiological yield from 45.8% to 80.4% [25]. Assuming a more conservative improvement, with an odds ratio of 1.25, this would require a sample of 194 patients for a power of 80%.

In urinary tract infection, a systematic review has found mNGS improves microbiological diagnosis when urine is collected from 61.7% to 90.5% [26]. Again, at a conservative estimate for effect with an OR of 1.25, this would require a sample of 108 patients to give a power of 80%.

Previous exploratory work within our lab has suggested neutrophils from older patients with CAP have 62% reduction in chemotaxis compared to age matched controls, with a SD 0.079 for CAP and 0.135 for controls, and a resulting sample size of 8 CAP patients and 8 controls. While this impact may be less notable among a cohort that includes younger adults, we would still expect the sample sizes outlined above to be more than adequate.

Allowing for study withdrawal rates of 10% we will therefore look to recruit a total sample of minimum 340 patients with infection, (230 with pneumonia, and 120 with UTI) and 100 age and co-morbidity matched controls without infection.

## Sample collection

All participants will have blood taken. Other samples will be taken depending on the suspected source of infection (see Table 3). Where samples are non-invasive (such as a spontaneous sputum sample, urine sample, faecal sample, saliva sample, skin swab) or minimally invasive (a blood test, a throat or nose swab), these will be taken as research investigations, and will not be part of routine clinical care.

Where sample collection is invasive (bronchoalveolar lavage or cerebral spinal fluid), samples will only be collected as part of routine care. Here, research samples would only be collected after clinical samples had been taken, with the agreement of the patient or personal/professional consultee.

**Table 3. Samples to be collected.**

| Suspected source of infection | Sample name | Proposed volume for collection |
|---|---|---|
| **All** | Blood | Up to 50 mL |
| **Respiratory** | Sputum | Up to 300 mg |
| | Saliva | Up to 5 ml |
| | Bronchoalveolar lavage fluid | Up to 60 ml |
| | Tracheal aspirates via endotracheal tube | Up to 30 ml |
| **Urinary tract** | Urine (MSU or CSU if catheterised patient). | Up to 10 ml |
| **Cerebral/ spinal** | Cerebrospinal fluid | Up to 3 ml |
| **Gastrointestinal** | Faeces | Up to 300 mg |
| **Skin/ mucous membrane** | Swab | Sterile moist cotton bud tip |
| **Skin** | Tape strip | Sterile tape strips |

**Sample transport and storage.** Samples will be transferred immediately after collection by a member of the research team in sealed containers labelled with the patient's study code and date of collection. Samples will be stored in a coded format in alarmed -80˚C freezers. A key, linking stored samples to clinical information will be maintained by the chief investigator, protected by specific log ins and auditable in terms of access. Where the type of suspected infection requires it, samples will be processed within a CL3 laboratory environment, in keeping with UK law on sample processing.

## Sample analysis

**Pathogen and AMR identification through genetic analysis.** It is important to note that genetic studies will only be performed on the pathogens and not on the host cells. DNA sequencing approaches may also incidentally sequence human DNA material present in samples. This will not be targeted by the methodology and sequences generated that map to the human genome will be removed by filtering and permanently deleted.

Whole genome sequencing will be performed on infected samples by using the Oxford Nanopore GridIon platform *(Nanopore Technologies, Oxford, UK)*.

Species identification will be provided by the MicrobesNG's analysis pipeline *(MicrobesNG, Birmingham UK)*, which uses Kraken [27]. Species will be further confirmed and sequence type designations conferred using mlst (version 2.15; Seeman T, https://github.com/tseemann/mlst) [28], and species-specific phylogenies, showing all members of the same species clustering together. We will assemble pathogen genomes using Trycycler [29] and annotate using Prokka [30].

AMR genes will be detected using ABRicate (version 8.7) [31]; using the National Center for Biotechnology Information (NCBI) AMRFinderPlus database [32]. Phylogenies will be initially reconstructed using Mashtree (version 2) [33], and visually inspected for clusters and AMR genes of importance using Phandango [34]. Isolates of the same species and sequence type or isolates appearing to cluster together on visual inspection of a phylogenetic tree will be assessed using Snippy (v 4.3.6) [35] on standard settings, using annotated assemblies as references to allow inference on SNP location and amino acid change.

Annotated assemblies of clustered isolates will be inspected in Artemis [36], and searches performed using the Basic Local Alignment Search Tool (BLAST) on the National Center for Biotechnology Information database [37] to elucidate the genes in which SNPs are observed.

**Immune cell isolation.** Whole blood will be processed using discontinuous Percoll density gradients as described previously [38]. This will result in separate layers of neutrophils, monocytes and plasma. Monocyte-Derived-Macrophages (MDMs) will be generated from monocytes after adherence, followed by culture in 2ng/ml GM-CSF for 12 days to generate MDM as described previously [39]. Typically, isolation by this method will yield a neutrophil population of >97% purity, >95% viability, and a monocyte population >95% purity, >95% viability. Total yield of cells and baseline viability are anticipated to vary based on severity of participant illness.

Plasma and serum samples will be collected from each participant, and aliqotedaliq into pre-labelled 1mL tubes for storage at -80˚C. These samples will be used to assess soluble markers of inflammation by immunoassay, in pooled plasma experiments to induce dysfunction in neutrophils from healthy controls. Peripheral blood mononuclear cells will be stored in liquid nitrogen for further phenotypic analysis.

**Immune cell function.** Isolated neutrophils and MDMs generated from patient monocytes will be used to characterise cellular phenotype and effector functions. Due to variable cell yield form patient samples not all assays will be performed for each participant.

The bioenergetics of the isolated neutrophils [40] and MDMs will be investigated by Seahorse XF analyser (Agilent Technologies, UK). Lysates of isolated and permeabilised immune cells will be prepared and stored at -80˚C for use in PCR and Western Blot analysis.

Neutrophil and MDM phenotype and function will be assessed as appropriate to cell type; Phenotype: flow cytometry; Migration: insall chamber and transwell assay [40]; Phagocytosis: labelled heat-inactivated *Streptococcus pneumoniae*, *Haemophilus influenzae*, and *Escherichia coli* (opsonised and unopsonised) [38]; Reactive oxygen species: CellRox assay; Neutrophil extracellular traps (NETosis): sytox green and Western blot, probing for citrullinated Histone 3.

**Transcriptomic and proteomic profiling.** Total protein and RNA will be extracted from neutrophils and MDMs by trizol extraction. RNA quality will be measured using the Agilent 2200 Tapestation (Agilent Technologies, UK). The polyA+ mRNA fractionation will be isolated and cDNA libraries prepared using the QuantSeq 3'mRNA-Seq Library Prep Kit (Lexogen, Austria). Samples will then be subjected to 75bp, paired-end sequencing using Illumina 2500 sequencing machine (Illumina Inc. San Diego, Ca, USA). Paired end reads will be aligned to the human reference genome (hg38) using STAR [41] and differential gene expression will be determined using DESeq2 [42].

Proteomic analysis will be carried out using advanced Liquid chromatography–high resolution mass spectrometry (LC-HRMS) techniques to examine changes in host immune cell function compared to matched, non-infected controls. Differentially represented proteins will be identified from the resulting HRMS data (fold change $> 1.5$ and $< 0.66$).

Data from both transcriptomics and proteomics analysis will be integrated for pathway analysis.

**Data sharing of pathogens to UK-HAS.** The UK Health Security Agency (UK-HSA) has a programme to track AMR and changes to the genetic signals of pathogens (including bacteria and viruses) known as MScape. The UK-HSA are asking laboratories performing genetic studies on pathogens to share data of the pathogen only (with no clinical or demographic data shared about the host). To contribute to this effort, we will send genetic codes of pathogens to UK-HSA but will not share data about the participants with UK-HSA.

**Data sharing of pathogens to patients and care teams.** Throughout this study, results from the novel, pathogen-genetics approach will be compared to the current gold standard results from NHS laboratories. We will compare the microbes identified and the time taken to identify the microbes using both the novel and gold standard approach. In applying novel diagnostic techniques, it is possible that we will identify pathogens that were previously unidentified by standard clinical care. The techniques used are not validated for clinical diagnostic testing or reporting and do not produce clinically actionable results.

All results will be discussed with the clinical microbiology team at University Hospitals Birmingham NHS Foundation Trust and the microbiology team will decide whether further tests are needed and communicate this to the direct care team. If clinically important organisms are identified (eg HIV or viral hepatitis), the clinical team will perform current gold standard tests to confirm the diagnosis and then initiate appropriate care.

If high consequence infectious diseases (HCID) as defined by UK-HSA are identified [43], we will inform both the clinical care team and UK-HSA. Of note, the participant will be made aware of this and consent to this approach as part of study participation.

## Data management

Data collected will include demographics (including date of birth, weight, height, ethnicity, and admission diagnosis), physiological indices (including heart rate, blood pressure, respiratory rate, temperature and oxygen saturations), results of relevant radiological investigations,

microbiological investigations and haematological and biochemical measurements, collected as part of routine clinical care.

To assess longer term outcomes including death, readmission to hospital or recurrent suspected infections, shared primary and secondary care patient notes will be reviewed at 1 month, 3 months and 1 year.

Pseudonymised data from the study will be held on a bespoke database. The database will be stored securely on a password-protected computer at the University of Birmingham, on servers that are backed up daily.

Data generated as part of multiomic analysis will be uploaded in an appropriate database (such as European Genome-phenome Archive) to allow controlled access.

## Statistical analysis

Appropriate statistical analysis will be performed with R v4.2 (The R Foundation, Austria) and GraphPad Prism V9.0 or later (Dotmatics, USA). Chi-squared test (for categorical data), independent sample T-tests, or non-parametric equivalents will be used as appropriate, to test for statistical significance of observed differences between groups. Odds ratios will be calculated, and multiple regression analysis will assess the impact of patient demography, co-morbidities or pathogen on patient outcomes.

## Patient and public involvement

Patient and members of the public were involved in the study design and will support the dissemination of results. Patients and their carers prioritised three areas of focus; faster diagnostics to better target antibiotics, better tools to identify those at risk of deterioration during infections, and new approaches to treat infection which harness the body's immune system.

Moving forward, study delivery will include the development of a patient/public involvement group with demographics that reflects the diversity of Birmingham. Through workshops, we will explore perceptions of rapid, genomics informed treatment, and co-create lay resources explaining antibiotic stewardship using rapid diagnostics. We will assess the impact of our patient and public involvement using the Public Involvement Impact Assessment Framework (PiiAF) [44].

## Author Contributions

**Conceptualization:** Samuel Quarton, Kirsty McGee, Nicola Cumley, Mahboobeh Behruznia, Charlotte Jeff, Kylie Belchamber, Michael Cox, David Thickett, Aaron Scott, Dhruv Parekh, Alan McNally, Elizabeth Sapey.

**Methodology:** Samuel Quarton, Kirsty McGee, Nicola Cumley, Mahboobeh Behruznia, Charlotte Jeff, Kylie Belchamber, Michael Cox, David Thickett, Aaron Scott, Dhruv Parekh, Alan McNally, Elizabeth Sapey.

**Writing – original draft:** Samuel Quarton, Elizabeth Sapey.

**Writing – review & editing:** Kirsty McGee, Nicola Cumley, Mahboobeh Behruznia, Charlotte Jeff, Kylie Belchamber, Michael Cox, David Thickett, Aaron Scott, Dhruv Parekh, Alan McNally.

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
