## [Decision Letter · Decision Letter 0]

9 Jan 2024

PONE-D-23-36821Towards personalised anti-microbial and immune approaches to infections in acute care. Can real-time genomic-informed diagnosis of pathogens, and immune-focused therapies improve outcomes for patients?  An observational, experimental study protocol.PLOS ONE

Dear Dr. Quarton,

Thank you for submitting your manuscript to PLOS ONE. After careful consideration, we feel that it has merit but does not fully meet PLOS ONE’s publication criteria as it currently stands. Therefore, we invite you to submit a revised version of the manuscript that addresses the points raised during the review process.

We look forward to receiving your revised manuscript.

Kind regards,

Benjamin M. Liu, MBBS, PhD, D(ABMM), MB(ASCP)

Academic Editor

PLOS ONE

Journal Requirements:

"This study is funded by the National Institute for Health and Care Research (NIHR) Birmingham Biomedical Research Centre (BRC). The views expressed are those of the author(s) and not necessarily those of the NIHR or the Department of Health and Social Care.

It is also supported by PIONEER, the Health Data Research UK (HDR-UK) Health Data Research Hub in acute care. HDR-UK is an initiative funded by the UK Research and Innovation, Department of Health and Social Care (England) and the devolved administrations, and leading medical research charities."

**Additional Editor Comments:**

Comments from Academic Editor:

1. The authors state "there is currently significant delay between sample collection and report read out, with average turnaround times for results being three days (13), due to the time taken to complete standard microbial culture and sensitivity assays." It is inappropriate to over generalize that turnaround time are 3 days as some assays provide quick turnaround. Also to make such statement the authors should define the clinical settings, e.g., inpatient and outpatient. The author's statement may be misleading. In addition, the turnaround time and delay in reporting is due to multiple factors, including but not limited to the medical systems and their efficiencies, clinical team, and labs. The authors statement should be modified to give the audience a whole picture.

2. The authors state "Modern genomic and molecular-based methods have the potential to rapidly identify not only the exact type of pathogen, but also genetic mutations which might affect virulence and antibiotic resistance (14)." The authors are suggested to add pros and cons of molecular testings. Besides, more references should be added. For instance, the following references may be relevant, it is optional to cite them. They can support the advantages of molecular assays and facilitate discussion on pros and cons of these tests comparing with other traditional tests.

Laboratory diagnosis of CNS infections in children due to emerging and re-emerging neurotropic viruses. Pediatr Res. 2023 Dec 2. doi: 10.1038/s41390-023-02930-6. Epub ahead of print. PMID: 38042947.

Novel HBV recombinants between genotypes B and C in 3'-terminal reverse transcriptase (RT) sequences are associated with enhanced viral DNA load, higher RT point mutation rates and place of birth among Chinese patients. Infect Genet Evol. 2018 Jan;57:26-35. doi: 10.1016/j.meegid.2017.10.023. Epub 2017 Oct 27. PMID: 29111272.

Reviewers' comments:

Reviewer's Responses to Questions

**Comments to the Author**

1. Does the manuscript provide a valid rationale for the proposed study, with clearly identified and justified research questions?

Reviewer #1: Yes

2. Is the protocol technically sound and planned in a manner that will lead to a meaningful outcome and allow testing the stated hypotheses?

Reviewer #1: Yes

3. Is the methodology feasible and described in sufficient detail to allow the work to be replicable?

Reviewer #1: Yes

4. Have the authors described where all data underlying the findings will be made available when the study is complete?

Reviewer #1: Yes

5. Is the manuscript presented in an intelligible fashion and written in standard English?

Reviewer #1: Yes

6. Review Comments to the Author

You may also provide optional suggestions and comments to authors that they might find helpful in planning their study.

Reviewer #1: I recommend the study protocol but the need the minor changes related to the study

Mention the proper heading for the following such as

Builds on previous work

Differs from previous work

7. PLOS authors have the option to publish the peer review history of their article (what does this mean?). If published, this will include your full peer review and any attached files.

Reviewer #1: **Yes: **Qaisar Ali

---

## [Author Response · Author response to Decision Letter 0]

22 Jan 2024

Thanks for your constructive and helpful comments. 

We have tried to address the issues raised by modifying our statement on turnaround times, expanding on the pros and cons of molecular testing, and including a summary of how the proposed study builds on and differs from previous work. We have provided the full detail of these responses within our 'Response to Reviewers' letter.

---

## [Editor Report · Decision Letter 1]

25 Jan 2024

Towards personalised anti-microbial and immune approaches to infections in acute care. Can real-time genomic-informed diagnosis of pathogens, and immune-focused therapies improve outcomes for patients?  An observational, experimental study protocol.

PONE-D-23-36821R1

Dear Dr. Quarton,

We’re pleased to inform you that your manuscript has been judged scientifically suitable for publication and will be formally accepted for publication once it meets all outstanding technical requirements.

Kind regards,

Benjamin M. Liu, MBBS, PhD, D(ABMM), MB(ASCP)

Academic Editor

PLOS ONE
---

## [Editor Report · Acceptance letter]

20 Mar 2024

PONE-D-23-36821R1 

PLOS ONE

Dear Dr. Quarton, 

I'm pleased to inform you that your manuscript has been deemed suitable for publication in PLOS ONE. Congratulations! Your manuscript is now being handed over to our production team.

Kind regards, 

on behalf of

Dr. Benjamin M. Liu 

Academic Editor

PLOS ONE